# Electrical Properties of In Situ Synthesized Ag-Graphene/Ni Composites

**DOI:** 10.3390/ma15186423

**Published:** 2022-09-16

**Authors:** Jingqin Wang, Dekao Hu, Yancai Zhu, Peijian Guo

**Affiliations:** 1State Key Laboratory of Reliability and Intelligence of Electrical Equipment, Hebei University of Technology, Tianjin 300130, China; 2Tianjin Research Institution of Electric Science Co., Ltd., Tianjin 300180, China

**Keywords:** electrical contact materials, Ag-graphene/Ni composite, graphene, welding force, arc energy

## Abstract

Ag/Ni composite contact materials are widely used in low-voltage switches, appliances, instruments, and high-precision contacts due to their good electrical conductivity and processing properties. The addition of small amounts of additives can effectively improve the overall performance of Ag/Ni contact materials. Graphene has good applications in semiconductors, thermal materials, and metal matrix materials due to its good electrical and thermal conductivity and mechanical properties. In this paper, Ag-graphene composites with different added graphene contents were prepared by in situ synthesis of graphene oxide (GO) and AgNO_3_ by reduction at room temperature using ascorbic acid as a reducing agent. The Ag-graphene composites and nickel powder were ball-milled and mixed in a mass ratio of 85:15. The Ag-graphene/Ni was tested as an electrical contact material after the pressing, initial firing, repressing, and refiring processes. Its fusion welding force and arc energy were measured. The results show a 12% improvement in electrical conductivity with a graphene doping content of approximately 0.3 wt% compared to undoped contacts, resulting in 33.8 IACS%. The average contact fusion welding force was 49.49 cN, with an average reduction in the fusion welding force of approximately 8.04%. The average arc ignition energy was approximately 176.77 mJ, with an average decrease of 13.06%. The trace addition of graphene can improve the overall performance of Ag/Ni contacts and can promote the application of graphene in electrical contact materials.

## 1. Introduction

Ag/Ni contact material is widely used in power relays, small circuit breakers, line protection switches, and high-precision contacts. Under some circumstances, it can take the position of a hazardous AgCdO contact material because of its benefits of superior processing flexibility, strong electrical and thermal conductivity, and low contact resistance.

Under high-current conditions, Ag/Ni contact materials can melt and weld, reducing the service life of the contacts. Fusion welding of contact materials often leads to safety accidents and reduces the reliability of electrical appliances. With the expansion of applications, new requirements have been placed on the resistance to fusion welding and arc erosion of Ag/Ni-based contact materials [1]. Many studies have shown that doping additives to Ag/Ni contacts can improve their overall performance, thereby increasing the resistance of the material to melt welding and arc erosion [2,3].

Graphene is a two-dimensional honeycomb monolayer sheet material composed of carbon atoms hybridized in sp2, which has attracted much attention due to its physical properties such as electrical, thermal and mechanical properties [4,5]. Graphene has unique two-dimensional properties and an associated energy band structure, where the conduction and valence bands intersect at the Dirac point [6]. Graphene has potential applications in catalysis [7], optoelectronics [8], solar cells [9], and other fields. Coated materials are a barrier, preventing all types of material surfaces from coming into contact with the outside world. The application of a coating material enhances the structural function to enable the material to be used in a particular environment [10,11,12]. The deposition of commercially available Ni50.8Ti (at.%) powder onto stainless steel substrates was investigated by De Crescenzo et al. using the high-speed oxy-fuel thermal spray technique, and the results showed that the greatest coating density was achieved with short sprays and that the greater the ratio of paraffin to oxygen, the greater the adhesion. Good adhesion conditions and good coating quality were achieved with the HVOF technique. Good technology facilitates the bond strength between the material interfaces [13]. Graphene and metals often form coatings or thin films that exhibit some excellent properties, leading to more potential applications. Chen C. et al. sintered Ag particles on bare copper substrates coated with a single layer of graphene, a technique that provides better interfacial oxide protection and mechanical reliability. This research could improve the lifetime of sintered Ag on copper-based substrates in power electronic packaging applications [14].

Graphene can also be used as an additive to strengthen metal matrices that are nickel-based, copper-based, and aluminum-based, and can improve the overall electrical conductivity or mechanical properties of composite materials [15,16,17]. Therefore, we used graphene as an additive to improve the electrical properties of Ag/Ni contact materials.

Ag-graphene composites are generally synthesized in situ from graphene oxide (GO) and metal salt solutions. the substrate of GO contains a large number of hydroxyl, epoxide, carbonyl, and carboxyl groups [18], which allows metal nanoparticles to bind to graphene by physical adsorption, charge transfer, and electrostatic bonding. The carboxylic acid group on the GO substrate can produce binding interactions with silver ions. GO removes the oxidation-containing functional groups in the presence of a reducing agent. It combines with silver to produce silver-reduced graphene oxide (Ag-rGO) composites. Reducing agents are generally used, such as borohydride [19] and hydrazine hydrate [20], but such reducing agents can be hazardous. Harmless reducing agents such as glucose [21] and sodium citrate [22] have also been used to reduce GO. Merino et al. investigated the deoxidation efficiency of vitamin C on GO at 95 °C and showed that vitamin C can compete with the prevalent but dangerous reducing agent hydrazine used in the production of graphene [23]. Gao et al. prepared a paper-like Ag-graphene composite film using ascorbic acid as the reducing agent and showed that the mechanical properties of the resulting film were improved with an increase in tensile strength and Young’s modulus of 82% and 136%, respectively [24]. Hui et al. performed in situ sonications of a mixture of AgNO_3_ and GO solutions at room temperature with vitamin C as the reducing agent to control silver nanoparticle-decorated graphene oxide (AgNP-GO) composites in terms of size [25]. Yang et al. prepared graphene with an added content of 1 wt% using a chemical exfoliation method. The prepared Ag-graphene composites showed a significant increase in electrical conductivity by 11% and microhardness by 42% compared to pure Ag [26]. Hao et al. used a chemical reduction method to introduce silver particles into graphene nanosheets and prepared silver-doped graphene nanosheets for silver-based composites by powder metallurgy. The results showed that the 0.5 wt% graphene doping content of the graphene-silver material increased the hardness of the silver-based composites by 35.1% and the electrical conductivity up to 98.62% IACS compared to pure Ag [27]. Graphene production is currently increasing every year and there are many areas of extended research on graphene applications. Therefore, we attempted to composite Ag-graphene with a nickel material for application in electrical switches to improve the material’s performance and obtain a graphene and Ag/Ni composite material with good resistance to fusion welding and less susceptible to arcing.

Both metals, Ag and Ni, have a low solubility in the solid state and cannot combine into a single liquid phase at temperatures higher than the Ni melting point (1453 °C) [28]. Therefore, when preparing Ag/Ni contact materials, powder metallurgy is typically used as the primary preparation technique.

We chose safe and green ascorbic acid (vitamin C) as the reducing agent to co-reduce the AgNO_3_ and GO suspensions, the resulting material was subjected to X-ray diffraction (XRD), and all GO and AgNO_3_ were found to be reduced. Nickel powder and Ag-graphene were combined in a ball mill for two hours. The samples’ electrical contact qualities were evaluated following the powder’s pressing, initial firing, repressing, and refiring. Measurements were made of its various characteristics, including fusion welding force, contact angle, and arc energy.

## 2. Materials and Methods

### 2.1. Materials

Graphene oxide (GO) powder was prepared using the improved Hummers method from Nanjing XF Nano Company (Nanjing, China). It has a thickness of 0.8–1.2 nm and a flake diameter of 0.5–5 μm. AgNO_3_ was obtained from Tianjin Damao and polyvinylpyrrolidone (PVP) from Sinopharm (Beijing, China), and all chemicals were of analytical grade or above.

The X-ray diffractometer (XRD) was a Rigaku SmartLab model made in Japan, with a scan range setting of 6° to 90° and a scan speed of 5°/min.

We chose the JF04D, an electrical contact material test system developed by the Kunming Institute of Precious Metals, to test the samples, with a protection voltage setting of ±40 V, a test voltage of 24 V DC, and a current of 15 A DC.

### 2.2. Methods

Three portions of GO powder, each weighing 5 mg, 10 mg, and 20 mg, were sonicated for 30 min in a 100 mL solution of deionized water. We weighed three portions of 0.2 mol/L AgNO_3_ in 80 mL and combined them with GO solution after adding 1 mg of polyvinylpyrrolidone (PVP) for dispersion. At room temperature, 500 mL of 0.3 mol/L ascorbic acid solution was slowly added to reduce Ag and GO together, resulting in a white precipitate.

The resulting precipitated powder was washed several times with deionized water and ethanol, filtered, dried, weighed, and mixed with Ni powder to produce an Ag-graphene powder with a weight ratio to Ni powder of 85:15. A set of Ag/Ni powders without graphene was also prepared. The powders were mixed in a ball mill for 2 h and then pressurised at 800 MPa to produce a cylinder with a diameter of 10 mm. The experimental procedure is shown in Figure 1. The initial sintering was held at 600 °C for 1 h in a tube furnace under evacuation, cooled and repressurized, and sintered at 700 °C after pressing, resulting in a blocky cylinder with a 10 mm diameter and 2.5 mm height. We tested its hardness and electrical conductivity.

We performed 50,000 electrical contact tests on the sample using the JF04D electrical contact material test system at a DC voltage of 24 V and a DC current of 15 A. Its welding force and arc energy were measured. We prepared two sets of samples by the same method under the same experimental conditions. Although the mean and variance of the parameters varied slightly, the trends were generally consistent with the results.

## 3. Results and Discussion

### 3.1. Conductivity and Hardness Measurement

The conductivity of the Ag-graphene/Ni composites was tested using a Sigma-Scope SMP10 metal conductivity tester. The Vickers hardness was obtained by pressing a diamond into the diamond shape of the sample surface and measuring the diagonal of its indentation. The results of the measurements are shown in Table 1.

The hardness and conductivity behaved differently depending on the amount of graphene added. The addition of small amounts of graphene improves the overall performance of the material, with the conductivity showing a tendency to increase and then decrease as the amount added increases. At a doping content of approximately 0.3 wt% (10 mg), the sample showed the best conductivity, with a result of 33.8 IACS%, a 12% increase in conductivity compared to the undoped sample. The conductivity of 33.8 %IACS is slightly higher than that of silver-nickel oxide at 28.7 %IACS, which was prepared using the same process by Tianfu G et al. [29]. Graphene has a rough, folded surface. The graphene’s folded surface can effectively fix the surrounding metal material and enhance the metal bonding, making the metal denser. Ball milling and sintering can change the work function of the material and reduce contact resistance. Therefore, there may be a slight increase in conductivity. When the addition is high, graphene will agglomerate to produce gaps as impurity phases, which will increase the scattering of carriers and make the conductivity decrease.

As the graphene content increased, the hardness showed a gradual increase. At a doping content of 0.6 wt% (20 mg), the Ag-graphene/Ni material showed a maximum hardness of 118.48 Hv, an increase of 3.9% compared to the unaddressed Ag/Ni material. At a doping level of 0.3 wt% (10 mg), there is little change in hardness. The hardness is higher than the 102 Hv of AgNi15 prepared by h et al. [30]. The upper and lower surfaces of graphene are in full contact with the metal. A small addition amount may have a small effect on the hardness, and the hardness only shows a gradual increase when the addition amount reaches a certain level. Hard Ag/Ni contact materials are more durable and do not deform greatly under mechanical and electrodynamic forces. The high hardness allows the contact material to be more wear-resistant, enhancing the life of the contact and making it more reliable and stable in its working condition.

### 3.2. X-ray Diffraction

The samples before and after the reaction of graphene with AgNO_3_ were analyzed by XRD, as shown in Figure 2. From the results, it can be seen that the unreduced GO powder in Figure 2 (a) shows a C(001) sharp peak around 2θ = 9.2°. After reduction, the powder appears in (b) with a small C(002) peak at around 2θ = 25°, which indicates that the oxidation group of GO is reduced, and Ag(111) (200), (220), (311), and (222) crystalline surfaces also appear at 2θ = 38.2°, 44.4°, 64.6°, 77.6°, and 81.7°, which indicates that silver nitrate is also reduced to silver. This is in good agreement with the silver-based graphene material reported by Dai et al. [31], indicating that graphene oxide was reduced to graphene and that Ag singlet powder appeared.

### 3.3. Contact Angle Measurement

Using a cylindrical material with a 10 mm diameter and 1 mm height pressed from nickel powder as a substrate, pure silver powder and Ag-graphene composite powder were selected to be placed on the substrate. In a tube furnace under a vacuum, the silver was heated above its melting point (942 °C); we heated it to 1050 °C and held it for 30 min to melt the pure silver powder and Ag-graphene composite powder onto the nickel block substrate. After the samples had cooled, a contact angle meter was used to compare the difference between the added and unadded graphene.

The wetting angle is determined by the molecular interactions between the liquid and the solid. The smaller the contact angle, the greater the wettability. As the contact angle approaches 180 degrees, wettability decreases. Strong adhesion with weak cohesion produces very low contact angles and almost complete wetting. The electric contact material produces a certain melting precipitation effect under the high temperature of the electric arc. That is, at high temperatures Ni particles dissolve in the silver melt, and when the temperature drops and the arc is extinguished, Ni particles reprecipitate to exist in the Ag matrix. The repeated action of the arc, with Ni particles constantly dissolving and precipitating, will enhance the Ag and Ni bonding strength. When the contact angle is smaller, the adhesion between Ag and Ni increases at high temperatures, and Ag and Ni bond better, which is less likely to produce material spatter and reduce material loss. As shown in Figure 3, when (a) is the contact angle of Ag and Ni without the addition of graphene, the average is 97.75°, and (b) is the contact angle of the Ag-graphene composite powder with Ni, where the average is 71.55°. It can be seen that with the addition of graphene, the wetting angle becomes significantly lower, improving the wettability between silver and nickel, and, therefore, enhancing the viscosity of the melt pool at high temperatures and reducing the spatter loss that occurs due to arc erosion.

### 3.4. Welding Force

We used the electrical contact material test system, type JF04D, to test the samples. We set the protection voltage to ±40 V, the experimental current to 15 A DC, the voltage to 24 V DC, the distance between contacts to 2 mm and the contact pressure to 86 cN. We measured the arc energy and the welding force of the material. We set the number of experiments on the sample to 50,000.

The lower the welding force, the better the resistance to fusion welding, and the less likely it is that the contacts will stick together when in contact under operating conditions. Figure 4a–d shows the results of the welding force of AgNi15 contacts with different doping-quality graphene content. A statistical analysis of the results is shown in Table 2.

The welding force of the undoped graphene contact material (sample d) generally fluctuated considerably. This indicates that the undoped Ag/Ni contacts are relatively unstable. The contacts with a graphene doping of approximately 0.3 wt% (sample b) had the lowest welding force with an average value of approximately 49.49 cN. Compared to the non-graphene-doped contacts, the contacts with a doping content of 0.3 wt% (sample b) had an average reduction in welding force of approximately 8.04%.

The lower variance of the welding force distribution indicates that the contacts are less prone to sudden changes in the working condition, the welding force curve is flatter, and the working condition is more stable. Of the four groups of samples, the contact with 0.15 wt% doping has the lowest variance and is more stable in operation, but has a higher welding force. The variance of the 0.3 wt% doped contacts was 18.7, slightly weaker than the 0.15 wt% doped contacts.

The results show that at a graphene doping level of approximately 0.3 wt%, the melt resistance of the contact material is improved, the welding force is reduced by 8.04%, and the stability in the operating condition is improved. A small amount of graphene doping can effectively reduce the welding force of silver-nickel contacts and improve the lifetime of silver-nickel contacts during operation.

### 3.5. Arc Energy

During the opening and closing of the contacts, an arc or other discharge phenomenon is generated between the contacts. The high temperature of the arc can cause the material on the surface of the contact to melt and splash, causing damage to the contact surface and affecting the performance of the contact. The results of the change in combustion arc energy are shown in Figure 5 and a statistical analysis of the four samples is shown in Table 3.

Among the four sets of results, the graphene-doped samples a, b, and c had significantly lower ignition arc energy than the non-graphene-doped sample d. The curve of the ignition arc energy of the non-graphene-doped contacts was flat and stable, but the ignition arc energy was higher. The lowest arc energy was obtained for the graphene doping of about 0.3 wt% (sample b) with an average value of about 176.77 mJ. The burning arc energy is lower than the 200 mJ of AgNi15 prepared by Chen-Li et al. [32]. The average arc energy decreased by 13.06% compared to that of the undoped sample d. The results indicate that graphene doped at approximately 0.3 wt% can reduce the arc ignition energy of the material the best and can reduce the arc burn on the contact.

## 4. Conclusions

In this paper, we used a chemical in situ method to prepare Ag-graphene composites using ascorbic acid as a non-toxic reducing agent. The Ag-graphene composites were then mixed with nickel powder to prepare a circular block material with a mass ratio of 85:15. After we performed the pressing, sintering, repressing, and resintering processes on the samples, we compared the performance of the contact material with the addition of different mass ratios of graphene. The results show that the trace addition of graphene can improve the overall performance of Ag/Ni contacts and can promote the application of graphene in electrical contact materials. The specific findings are as follows:(1)The addition of graphene resulted in an increase in conductivity, with a 12% increase in conductivity at 0.3%wt. The hardness of the samples also tended to increase with a higher content.(2)The addition of graphene can make the contact angle of the Ag/Ni contact smaller and enhance the wettability between silver and nickel. Under the action of a high-temperature arc, the Ag/Ni contacts will reduce the spattering effect.(3)The addition of graphene reduces the fusion welding force and reduces the occurrence of fusion welding in the contacts, which can improve the life of the contacts. A graphene addition of 0.3%wt reduces the melt welding force by 8.04%, which is the best result.(4)The lowest arc energy was achieved at a graphene doping level of approximately 0.3 wt%, with an average value of approximately 176.77 mJ. Compared to the contacts without graphene doping, the average arc energy decreased by 13.06%, reducing the arc burn on the contacts.

## Figures and Tables

**Figure 1 materials-15-06423-f001:**
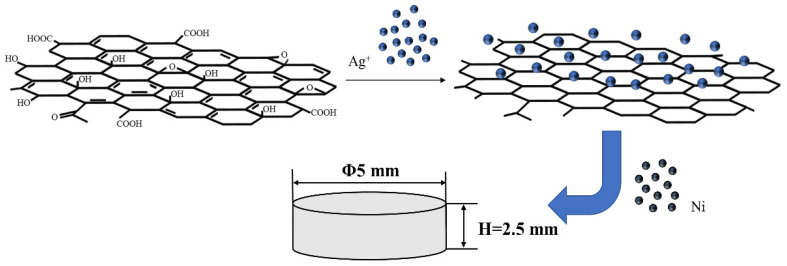
Schematic illustration of the synthesis of the Ag-graphene/Ni composite.

**Figure 2 materials-15-06423-f002:**
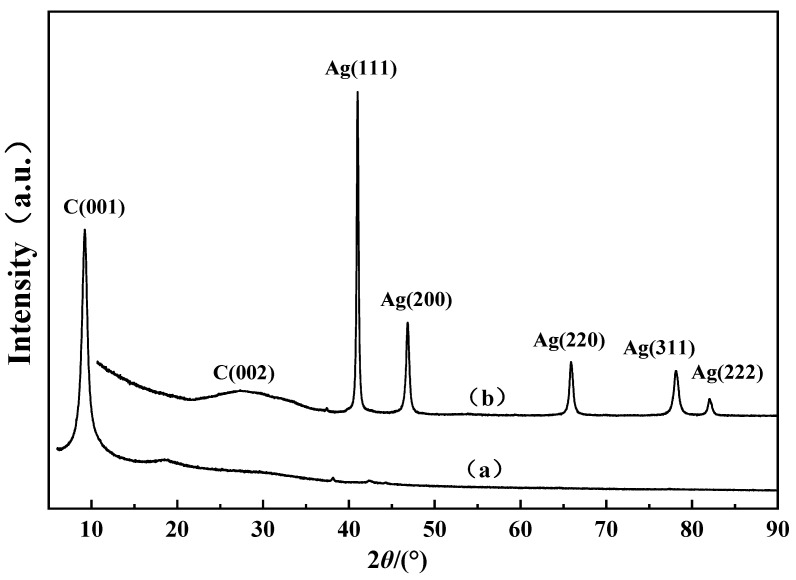
XRD patterns of (a) GO and (b) Ag-rGO.

**Figure 3 materials-15-06423-f003:**
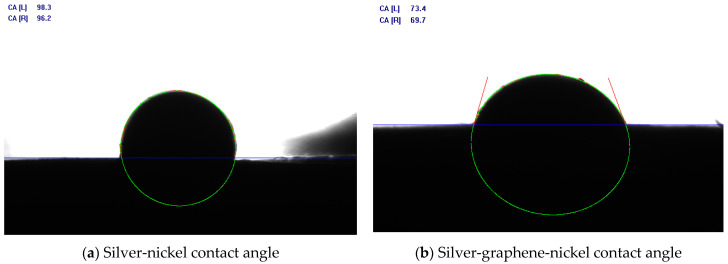
Silver-nickel contact angle before and after graphene addition.

**Figure 4 materials-15-06423-f004:**
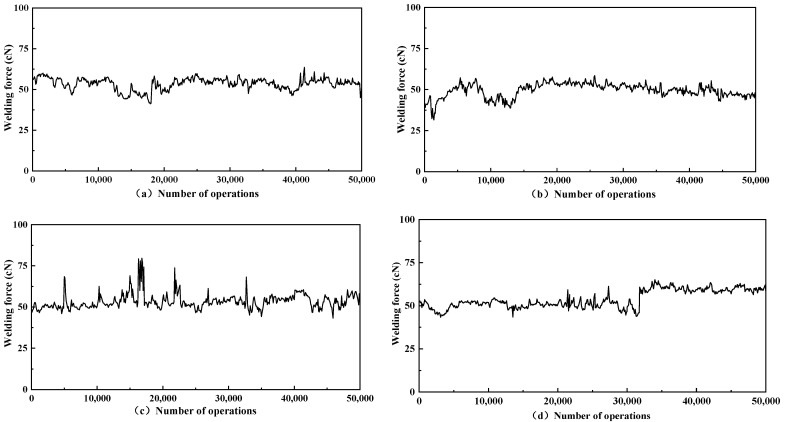
Ag-graphene/Ni contactor welding force: (**a**) AgNi15-5 mg GO, (**b**) AgNi15-10 mg GO, (**c**) AgNi15-20 mg GO, (**d**) AgNi15.

**Figure 5 materials-15-06423-f005:**
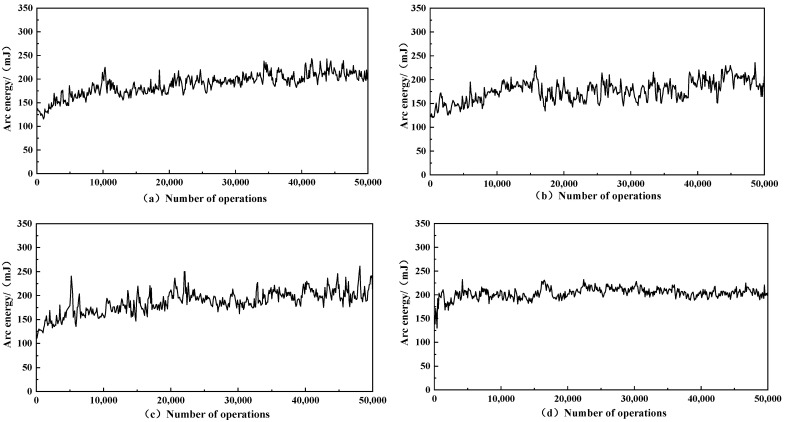
Ag-graphene/Ni contactor arc energy: (**a**) AgNi15-5 mg GO, (**b**) AgNi15-10 mg GO, (**c**) AgNi15-20 mg GO, (**d**) AgNi15.

**Table 1 materials-15-06423-t001:** Conductivity and hardness of the Ag-graphene/Ni.

Samples	Sample Name	Conductivity/(IACS%)	Hardness/HV
a	AgNi15-5 mg GO	31.3	112.70
b	AgNi15-10 mg GO	33.8	114.27
c	AgNi15-20 mg GO	27.4	118.48
d	AgNi15	30.0	114.05

**Table 2 materials-15-06423-t002:** Average welding force and variance of the Ag-graphene/Ni.

Samples	Sample Name	Average Welding Force/(cN)	Variance
a	AgNi15-5 mg GO	53.46	13.7
b	AgNi15-10 mg GO	49.49	18.7
c	AgNi15-20 mg GO	53.37	22.0
d	AgNi15	53.82	26.5

**Table 3 materials-15-06423-t003:** Average arc energy and variance of the Ag-graphene/Ni.

Samples	Sample Name	Average Arc Energy/(mJ)	Variance
a	AgNi15-5 mg GO	189.10	544.5
b	AgNi15-10 mg GO	176.77	486.4
c	AgNi15-20 mg GO	186.96	575.6
d	AgNi15	203.33	128.9

## Data Availability

The data used to support the findings of this study are available from the corresponding author upon request.

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
