# Peer review of "Electrical Properties of In Situ Synthesized Ag-Graphene/Ni Composites"

_materials, 2022, doi:10.3390/ma15186423_

Round 1

Reviewer 1 Report

General comment:

The paper deals with investigations on Ag-graphene composites with different added graphene contents. The materials was prepared by in situ synthesis of graphene oxide and AgNO3 by reduction at room temperature using ascorbic acid as reducing agent.

The manuscript is suitable to be published in this journal, however some point should be addressed before publication.

Some minor language mistakes are present that should anyway be corrected.

1. Introduction

Please, improve the literature overview on coating characterization and deposition. Please consider the following papers:

o   Development and characterization of high performance Shape Memory Alloy coatings for structural aerospace applications (2018) Materials, 11 (5), art. no. 832.

o   Low-pressure plasma-sprayed ZrO2-CaF2 composite coating for high temperature tribological applications, Surface and Coatings Technology, 137(1), 21-30.

o   Enhanced corrosion protection of NiTi orthopedic implants by highly crystalline hydroxyapatite deposited by spin coating: The importance of pre-treatment (2021) Materials Chemistry and Physics, 259, art. no. 124041.

o   Ni-Ti Shape Memory Alloy Coatings for Structural Applications: Optimization of HVOF Spraying Parameters (2018) Advances in Materials Science and Engineering, 2018, Article number 7867302.

2. Experimental Methods and Procedures

Please, specify if investigations were carried out in duplicate/triplicate etc.

Please, specify the operative parameters for the material preparation varied and their variation range.

3. Results and discussion

Please, improve comparison between your findings and literature data in terms of material characteristics and performance.

Author Response

Dear Editors and Reviewers,

Thanks very much for taking your time to review this manuscript. We really appreciate all your generous comments and suggestions.

  1. We have added a literature review on coating materials, specifically in rows 46-60, and have cited relevant papers.

(Coated materials are a barrier for all types of material surfaces to come into contact with the outside world. The application of a coating material enhances the structural function to enable the material to be used in a particular environment [10-12]. The deposition of commercially available Ni50.8Ti (at.%) powder onto stainless steel substrates was investigated by De Crescenzo et al. using the high-speed oxy-fuel thermal spray technique and the results showed that the greatest coating density was achieved with short sprays and that the greater the ratio of paraffin to oxygen, the greater the adhesion. Good adhesion conditions and good coating quality were achieved with the HVOF technique. Good technology facilitates the bond strength between the material interfaces [13]. Graphene and metals often form coatings or thin films that exhibit some excellent properties, leading to more potential applications. Chen C. et al. sintered Ag particles on bare copper substrates coated with a single layer of graphene, a technique that provides better interfacial oxide protection and mechanical reliability. This research could improve the lifetime of sintered Ag on copper-based substrates in power electronics packaging applications [14].)

  1. We prepared two samples by the same method and the experimental results are as follows. The experiment was verified twice under the same experimental conditions. Although the mean and variance of the parameters changed slightly, the trends were generally consistent with the findings of the first sample analysis. This proves the reproducibility of the experiment.

Samples

Sample name

Average arc energy/(mJ)

Variance

a

AgNi15-5 mg GO

196.26

160.38

b

AgNi15-10 mg GO

171.42

170.52

c

AgNi15-20 mg GO

180.62

458.65

d

AgNi15

201.97

977.39

Samples

Sample name

Average welding force/(cN)

Variance

a

AgNi15-5 mg GO

55.34

17.83

b

AgNi15-10 mg GO

50.52                                             

9.89

c

AgNi15-20 mg GO

54.57

13.61

d

AgNi15

54.78

61.16

  1. We refer to the conductivity hardness of Ag/Ni-oxide by the equivalent process of Tianfu G. et al. The conductivity is slightly higher than the 28.7% IACS of Ag/Ni-oxide by Tianfu G. et al. The ignition arc energy of our prepared Ag/Ni-graphene material is lower than that of AgNi15 by Li C. et al. by 200 mJ. The hardness is higher than that of the 102 HV prepared by H. Wang, and this difference may be due to the different preparation processes.The AgNi-graphene contacts perform well in terms of resistance to fusion welding. Because the preparation process of high-temperature extrusion is not available in the laboratory, some pores may exist on the surface of the material. Therefore, there is a gap with the industrial standard product.
  1. Li C, Ming X, Dekui N, Hongzhong C. Anti-welding Characteristic of AgRENi Electrical Contact Material on DC Condition. Precious Materials. 2008(03):6-10.
  2. H. Wang and H. Yuan. Investigation on the electrical properties of AgNi contact materials with various Ni content. 2017 IEEE Holm Conference on Electrical Contacts, 2017, pp. 221-224.
  3.  Tianfu G. Preparation and Properties of the Silver-Nickel Contact Material Addition with Metal Oxides[D]. Xi’an Polytechnic University,2016.

Reviewer 2 Report

The manuscript by Wang et al. described the syntheses and electrical property of Ag-graphene/Ni composites. Ag-graphene/Ni composites were made by first co-reducing AgNO3 and GO and then mixed with Ni. Their characterization results show that the trace addition of graphene can improve the overall performance of Ag/Ni contacts, including increased hardness, enhanced wettability, reduced fusion welding force and arc ignition energy. Overall it’s an interesting paper and I would recommend publication with minor revisions.

1.      While I understand the focus of the paper is to demonstrate their enhanced electrical properties, the authors should explain more about their material fabrication process. Why the Ni is added at a later stage, instead of co-reducing Ag, Ni salts and GO together? What’s the advantage and disadvantages of these composites? Would you expect a better performance?

2.      The hardness and conductivity behaved differently with the amount of graphene added. The authors should explain why the conductivity shows an increase and then decrease trend, and why the hardness continuously to increase.

Author Response

Dear Editors and Reviewers,

Thanks very much for taking your time to review this manuscript. We really appreciate all your generous comments and suggestions.

  1. The general mass ratios of Ag/Ni contact series materials are Ag/Ni (10), Ag/Ni (15), Ag/Ni (20), etc. We think it is not easy to control the mass ratio of Ag to Ni in the process of mixed Ag and Ni solutions for joint reduction. We think there is a little difficulty in the reduction process which metal rGO preferentially binds with preferentially, and if Ni is added at the beginning, it has magnetic properties and there is some difficulty in the subsequent characterization.Therefore, we sincerely accept your suggestion and hope to continue to study the silver-nickel co-reduction in depth in subsequent experiments.
  2. a). Due to different addition amounts, there may be effects on material grain size, crystal dislocations, and physicochemical properties. Therefore, the effects of different addition amounts need to be considered. Graphene has a rough pleated surface. The graphene folded surface can effectively fix the surrounding metal material and enhance the metal bonding, it makes the metal denser. Ball milling and sintering can change the work function of the material and reduce contact resistance. Therefore, there may be a slight increase in conductivity. When the addition is high, graphene will agglomerate to produce gaps as impurity phases, which will increase the scattering of carriers and make the conductivity decrease.     b).  Graphene has excellent mechanical properties and is a better material to be used as a reinforcing phase to improve the strength of the material. Both the upper and lower surfaces of graphene can be in full contact with the silver material. The extremely trace additions may have a small effect on the hardness, and only when the additions reach a certain level does the hardness show a gradual enhancement.